# Redirecting T Cells against Epstein–Barr Virus Infection and Associated Oncogenesis

**DOI:** 10.3390/cells9061400

**Published:** 2020-06-04

**Authors:** Christian Münz

**Affiliations:** Viral Immunobiology, Institute of Experimental Immunology, University of Zürich, 8057 Zürich, Switzerland; christian.muenz@uzh.ch

**Keywords:** T cell receptor, chimeric antigen receptor, adoptive T cell transfer, diffuse large B cell lymphoma, nasopharyngeal carcinoma, latent membrane protein, EBV nuclear antigen

## Abstract

The Epstein–Barr virus (EBV) is associated with lymphomas and carcinomas. For some of these, the adoptive transfer of EBV specific T cells has been therapeutically explored, with clinical success. In order to avoid naturally occurring EBV specific autologous T cell selection from every patient, the transgenic expression of latent and early lytic viral antigen specific T cell receptors (TCRs) to redirect T cells, to target the respective tumors, is being developed. Recent evidence suggests that not only TCRs against transforming latent EBV antigens, but also against early lytic viral gene products, might be protective for the control of EBV infection and associated oncogenesis. At the same time, these approaches might be more selective and cause less collateral damage than targeting general B cell markers with chimeric antigen receptors (CARs). Thus, EBV specific TCR transgenic T cells constitute a promising therapeutic strategy against EBV associated malignancies.

## 1. Introduction of EBV and Its Oncogenesis

The Epstein–Barr virus (EBV) was discovered in 1964, and was the first human tumor virus [1,2]. It is still, to date, the most potent pathogen to transform human B cells into immortalized lymphoblastoid cell lines (LCLs) in vitro [3]. Despite this high oncogenic potential and its classification as a WHO class I carcinogen [4,5,6], most adult humans carry EBV asymptomatically. Indeed, more than 95% of the human adult population is persistently infected with EBV, and the infection programs in healthy virus carriers are the same as have been found in EBV associated malignancies [7,8]. The default program of B cell infection by EBV is the growth transforming latency III, expressing six nuclear antigens (EBNAs) and two latent membrane proteins (LMPs), together with viral non-translated small RNAs (EBERs) and miRNAs (Figure 1). This viral gene expression pattern is also found in EBV associated post-transplant lymphoproliferative disease (PTLD), HIV associated immunoblastic lymphoma, some diffuse large B cell lymphomas (DLBCL) and LCLs [9]. It is thought to drive EBV infected naïve B cells, in which latency III is found in healthy EBV carriers [10], into differentiation to memory B cells, the reservoir of long-term viral persistence [11]. The next step after latency III in this differentiation path is thought to be the germinal center differentiation of B cells, and EBV reduces its latent gene transcription to EBNA1 and the two LMPs plus non-translated RNAs to facilitate the survival of infected B cells [12]. Indeed, this latency II program can be found in the germinal center B cells of healthy virus carriers. At this differentiation stage, uninfected B cells acquire somatic mutations to increase antigen affinity of their B cell receptor [13]. Unfortunately, the same mechanism also favors pro-oncogenic mutations like c-myc transloctions, and EBV associated Hodgkin’s and Burkitt’s lymphoma are thought to originate from this differentiation stage [14]. Hodgkin’s lymphoma expresses latency II, and in most Burkitt’s lymphomas, only EBNA1 is expressed as the sole viral protein. Via germinal center differentiation, EBV infected B cells can reach the memory B cell pool for long-term persistence. Persistence can also be reached without latency III, albeit less efficiently and probably via the direct infection of memory B cells [15]. In memory B cells, no viral proteins, but only non-translated RNAs are expressed, in so called latency 0 [11]. During their homeostatic proliferation, EBNA1 is transiently expressed in latency I that is also found in Burkitt’s lymphoma [16]. From latency 0 and I, the infectious particle producing lytic EBV replication can be induced upon plasma cell differentiation, presumably after B cell receptor engagement [17].

This lytic replication can then be amplified through lytic replication in mucosal epithelial cells for efficient viral shedding into the saliva that transmits EBV to new hosts [18]. EBV associated epithelial cell cancers are thought to originate from this infection, but carry abnormal latent EBV gene expression. For example, nasopharyngeal carcinoma (NPC) often expresses latency II and EBV associated gastric carcinoma sometimes even latency I [19,20]. Therefore, all oncogenic programs of EBV infection are present in healthy virus carriers. While PTLDs are often oligoclonal and primarily driven by EBV oncogenes, the EBV associated malignancies with latency I and II expression programs require additional somatic mutations and are therefore often monoclonal. Accordingly, PTLD development can be rapid, often occurring during the first year after organ transplantation, while Burkitt’s and Hodgkin’s lymphoma occur at later timepoints after transplantation [21]. In contrast, in HIV infected individuals, Burkitt’s and Hodgkin’s lymphoma emerge earlier than latency III immunoblastic lymphoma [22]. This is best explained by the inverse gradients of immune suppression that are observed after transplantation and HIV infection. While immune suppression is most pronounced early after transplantation and late in HIV infection when immunogenic EBV latency III lymphomas occur, Hodgkin’s and Burkitt’s lymphoma, with restricted viral antigen expression, occur late after transplantation and earlier in HIV infection in the presence of recovered or not yet sufficiently damaged immune responses. These considerations suggest that efficient immune responses seem to protect us from EBV associated lymphomas.

## 2. Immune Control of EBV

This immune control is mainly mediated by cytotoxic lymphocytes, including natural killer (NK) and CD8^+^ T cells that expand during symptomatic primary EBV infection, so called infectious mononucleosis (IM) [23]. Furthermore, the adoptive transfer of EBV specific T cell lines is used to treat some of the above mentioned EBV associated malignancies, including PTLD, Hodgkin’s lymphoma and NPC [24,25,26]. Cytotoxic lymphocytes, predominantly CD8^+^ T cells, are also identified as the cornerstones of EBV specific immune control by primary immunodeficiencies that predispose for EBV associated pathologies [27,28,29]. The underlying mutations affect genes that are either required for perforin mediated cytotoxicity, the co-stimulation of cytotoxic lymphocytes, T cell receptor signaling, generation of cytotoxic lymphocytes or their efficient expansion and survival. In contrast, loss of type I and II interferons or antibody responses does not seem to predispose for EBV associated pathologies [28]. Deficiencies in cytotoxic lymphocytes result in continuous immune stimulation without efficient immune control of EBV and manifest as IM, hemophagocytic lymphohistiocytosis, lymphoproliferative diseases and virus associated B cell lymphomas. In severe cases, only bone marrow transplantation provides a valid therapeutic option [29]. In order to understand the functions of the cytotoxic lymphocyte populations and of molecules that are identified by mutations in primary immunodeficiencies during EBV specific immune control, mice with reconstituted human immune systems (HIS mice) are explored [30]. For this purpose, mice that lack murine lymphocytes due to genetic deficiencies are neonatally reconstituted with human CD34^+^ hematopoietic progenitor cells and their human immune system components that develop after three months allow EBV infection, associated oncogenesis and cell-mediated immune control [30]. In these mice, the contribution of cytotoxic lymphocytes to EBV specific immune control has been addressed. Protection by CD8^+^ and CD4^+^ T cells, NK cells, NKT cells and γδ T cells has been demonstrated [31,32,33,34,35,36,37,38]. In addition, the pharmacological inhibition of T cell function by FK506 (tacrolimus), that increases the risk for PTLD development in transplant patients elevates EBV loads in infected HIS mice and virus associated lymphoma incidence [39]. Moreover, antibody mediated blocking of the co-receptors 2B4 or PD-1 on T cells also compromises EBV specific immune control in HIS mice [40,41]. Depending on the viral dose of infection, an IM like expansion of CD8^+^ T cells can be induced upon EBV infection, accumulating PD-1^+^ CD8^+^ T cells, with a germinal center homing phenotype reminiscent of tonsillar EBV specific CD8^+^ T cells in humans [41,42,43]. In good agreement with findings in primary immunodeficiency patients type I interferon and its main producers, plasmacytoid dendritic cells had little effect on EBV infection in HIS mice [44]. In combination, the above discussed evidence suggests that cytotoxic CD8^+^ T cells are the most suitable immune compartment to control persistent EBV infection long-term and to prevent virus associated malignancies.

## 3. Antigen Specificity of Protective T Cell Responses against EBV

EBV, however, encodes eight latent proteins and more than 80 lytic gene products, all of which are potential targets for such protective cytotoxic CD8^+^ T cell responses [45,46]. In order to reestablish immune control in patients with EBV associated tumors and protect vulnerable populations by prophylactic vaccination, a subset of these antigens has to be selected to isolate protective T cell clones or their T cell receptors for adoptive transfer into HLA matched donors, or to incorporate them into suitable vaccines. Originally, the respective studies primarily focused on the latent EBV antigens that contain the viral oncogenes and are expressed in the EBV transformed tumor cells. These studies revealed that healthy EBV carriers primarily recognize EBNA3A, 3B and 3C of latency III and to a lesser extent EBNA1, EBNA2 and LMP2, but rarely EBNA-LP and LMP1 [45]. In particular, EBNA2 specific CD8^+^ T cell responses can kill EBV infected B cells early after virus exposure, due to the early expression of this latent viral gene product [47]. The less frequent responses often depend on the presence of distinct major histocompatibility complex (MHC) class I molecules as the presenting molecules for the respective antigens [48]. In contrast, MHC class II restricted CD4^+^ T cells recognize a broad range of latent EBV antigens, with EBNA1 specific CD4^+^ T cells present in every healthy virus carrier [49,50]. All latent EBV antigen specific CD8^+^ and many CD4^+^ T cells directly recognize LCLs and can kill them, suggesting protective functions against EBV infection and its pathologies in vivo [51,52,53]. Among these T cell responses, EBNA1 and/or LMP1 and LMP2 specific T cells have been selectively isolated or expanded for adoptive transfer into patients with EBV associated malignancies, resulting in clinical benefits [26,54,55,56]. Moreover, EBNA1 incorporation into EBV derived virus like particles or recombinant adenoviruses and modified vaccinia virus Ankara (MVA) induced protection against infection challenge in mice with reconstituted human immune system components or tumor challenge in regular mice [57,58]. Therefore, EBNA1, LMP1 and LMP2 specific T cells are favored as protective entities targeting latent EBV proteins.

However, CD4^+^ and CD8^+^ T cells in even higher frequencies than those specific for latent EBV antigens recognize lytic viral gene products [45]. For example, early lytic EBV antigen specific CD8^+^ T cells against single peptide epitopes can constitute up to 40% of all CD8^+^ T cells in the peripheral blood of IM patients [59]. For a long time, their protective function was in doubt, because their cytokines most likely cause IM symptoms [60]. However, recently it was realized that the two early lytic viral Bcl-2 homologues BHRF1 and BALF1 are immediately and transiently expressed early after EBV infection of B cells, and therefore could be targeted for immune control [61,62,63]. Moreover, early lytic EBV gene expression was found to contribute to virus associated tumor formation [7,64,65]. Indeed, such cells with early lytic EBV infection can be targeted with CD8^+^ T cells that are specific for the early lytic EBV antigen BMLF1 [66]. CD8^+^ T cells of healthy EBV carriers recognize preferentially early lytic EBV antigens and less late structural proteins [45]. CD4^+^ T cells recognize, again, a wider spectrum of lytic antigens including late viral gene products, and such late lytic EBV antigen specific CD4^+^ T cells have been found to inhibit EBV associated lymphoma formation in mice [67]. Therefore, also lytic EBV antigen specific T cells, especially CD8^+^ T cells that recognize early lytic viral gene products, should be considered for their protective function against infection and virus associated malignancies.

## 4. EBV Specific T Cell Receptors in Preclinical Studies

Instead of expanding EBV specific T cells, the receptors of these T cells (TCRs) have been explored to redirect CD4^+^ and CD8^+^ T cell populations, to recognize and eliminate EBV transformed B cells. Such TCRs have been generated from EBNA3A, EBNA3B, LMP1, LMP2, BRLF1 and BMLF1 specific CD8^+^ T cell clones [68,69,70,71] (Figure 2).

Unfortunately, T cell lines transgenic for these T cell receptors recognized autologous LCLs only weakly. This is in part due to limited latent EBV antigen expression in LCLs and even potent CD8^+^ T cell clones of healthy EBV carriers can only kill one third of LCLs, even when far outnumbering their target cells [72]. This low latent EBV antigen expression was proposed to lead only to the presentation of one copy of an EBNA3C peptide on every third LCL in culture [73]. Additionally, for lytic EBV antigen recognition, only a small percentage of LCLs are at any given timepoint in lytic replication, and can be targeted by lytic EBV antigen specific CD8^+^ T cells [66]. Several strategies have been explored to improve EBV positive tumor recognition by T cells with EBV specific TCR transgenes. The recombinant TCRs have been equipped with the co-stimulatory signaling domain of CD28 [69]. However, while the T cells with such an improved EBNA3B specific T cell receptor produced more IFN-**γ** in response to autologous LCLs, they did not kill these target cells more efficiently. Furthermore, different promotors were compared to drive the expression of a LMP2 specific T cell receptor [74]. While the V**β**6.7 promotor was identified to drive highest transgenic T cell receptor expression, the killing of autologous LCLs was not improved. Nevertheless, tumor progression by adoptive transfer of the CNE NPC line into nude mice could be significantly attenuated by TCR transgenic T cell transfer. As an additional approach, transgenic TCR expression in cord blood T cells was explored, in order to generate less differentiated EBV specific T cell lines [75]. While LMP2 specific TCR transgenic T cells with longer telomeres and early differentiation marker expression were obtained, these were not significantly better in killing EBV derived epitope presenting target cells. Therefore, it was reasoned that some of the transgenic TCR α and **β** chains might pair with the endogenous TCR and thereby dilute redirection for efficient LCL recognition. To counteract this, chimeric TCR were generated, containing mouse TCR constant regions and the variable domains of TCRs from EBV specific T cell clones [76]. Although autologous LCL recognition was not compared between LMP2 chimeric and entirely human TCRs, chimeric TCR transgenic T cells were able to efficiently reject LMP2 expressing tumor cells in immune compromised mice. CD8^+^ T cells with LMP2 and BMLF1 specific TCRs of similar design also efficiently expanded during EBV infection in HIS mice [41]. In these studies, stability of the murine TCR constant region pairing was improved by introducing an additional disulfide bond between the TCR α and **β** chain constant domains. A similar improvement was then chosen for another LMP2 specific codon optimized TCR [77]. Transgenic T cells with this receptor were able to kill up to half of co-incubated EBV positive NPC cells and to suppress LMP2 expressing tumor cell growth in immune compromised mice. More recently, LMP1 has also been targeted with TCR transgenic T cells [71]. Despite the two decades of development of EBV specific TCRs for transgenic expression, however, the killing of autologous LCLs remains a challenge and peaks also for this newest TCR at 20%. Nevertheless, the transfer of LMP1 specific TCR transgenic T cells doubles the survival of immune compromised mice that have been challenged with LMP1 expressing tumor cells. Thus, despite the seemingly low level of cytotoxicity against autologous tumor cells that, however, is also observed for EBV specific T cell clones, transgenic T cell therapy should be developed for clinical application against EBV associated malignancies.

## 5. EBV Specific Chimeric Antigen Receptors in Preclinical Studies

A higher level of in vitro cytotoxicity against B cell lymphomas can be achieved with T cells expressing chimeric antigen receptors (CARs). CAR transgenic T cells targeting CD19, CD20, CD22 and CD30 were able to kill around 50% of different B cell lymphomas in cell culture [78,79,80]. The respective CARs usually contain the variable region of antibodies against these B cell surface markers, a transmembrane domain and the CD3ζ T cell receptor signaling domain, with or without additional costimulatory signaling domains of CD28 or 4-1BB (CD137) or OX40 (CD134) [81]. These therapies have provided spectacular clinical results, often with 60% of B cell lymphoma and leukemia patients in complete remission after treatment [82,83]. However, due to the abundance of target cells, including both normal and malignant B cells, for the infused CAR transgenic T cells, significant side effects of cytokine release syndrome and neurotoxicity have also been observed [81]. Moreover, due to the persistence of B cell targeting CAR transgenic T cells, they could, with time, induce deficiencies in humoral immune responses when plasma cell-based antibody responses wane and would need to be replaced by memory B cells. These would then result in vulnerabilities predisposing for respiratory and gastrointestinal infections, as seen in combined variable immunodeficiency (CVID) patients [84]. In order to avoid these side effects, EBV, instead of B cell specific CARs, would be desirable to target EBV associated malignancies.

Along these lines, a CAR against LMP1 has been explored against NPC [85,86]. Although a substantial amount of LMP1 is actually located in endosomes and not accessible on the surface of EBV associated tumor cells for CAR recognition [87], NPC cells overexpressing LMP1 could be targeted by T cells transgenic for a LMP1 specific CAR [85,86]. Up to 70% of LMP1 overexpressing NPC cells could be killed in vitro and growth of these tumor cells could be arrested by the respective CAR expressing T cells upon transfer into immune compromised mice. A LMP1 specific CAR with CD3ζ and 4-1BB (CD137) signaling domains was most efficient in controlling tumor growth in vivo [85]. Unfortunately, these studies leave the question unanswered if EBV associated tumor cells with physiological levels of LMP1 expression can also be targeted by the viral antigen specific CAR. Nevertheless, both EBV surface antigen specific CARs and TCRs that are able to detect intracellular EBV antigen expression via MHC restriction should be further explored to treat EBV associated malignancies.

## 6. Clinical Development of EBV Specific TCRs and B Cell Specific CARs against EBV Associated Malignancies

Along these lines, more than 250 clinical trials with CARs targeting common B cell markers, primarily CD19, are currently registered with ClinicalTrials.gov (www.clinicaltrials.gov). Even among the trials that clearly indicate EBV associated malignancies as treatment targets, trials with CAR transgenic T cells outnumber trials with EBV specific TCR transgenic T cells (Table 1).

However, these clinical trials have not yet reported results. Nevertheless, two strategies emerge to provide TCR or CAR transgenic cellular products rapidly. This might be especially important for rapidly progressing EBV associated malignancies, including PTLDs [88]. On the one hand, leukapheresis of tumor patients is proposed, to harvest sufficient amounts of autologous T cells [89] for redirection with CARs or EBV specific TCRs (Figure 3). On the other hand, allogeneic T cells might serve as a platform for EBV^+^ tumor specific receptor expression, but these allogeneic T cells bear the danger of eliciting graft-versus-host-disease (GvHD). However, it was noted that EBV specific T cells carry only low alloreactivity [90], and are therefore favored as an allogeneic source in several clinical trials (Table 1). Along these lines, allogeneic EBV specific T cells could be banked and transduced with either HLA matched EBV specific TCRs, or tumor specific CARs, for infusion into patients with EBV associated malignancies. As can be seen from the registered clinical trials, these sources of T cells are mainly used for CAR transduction against EBV associated lymphomas, while EBV specific TCR transgenes are mainly explored for EBV associated carcinomas, mainly NPC, for which no specific CARs exist that would not cause severe side effects during treatment. Thus, both CAR and TCR transgenic T cell therapies are currently explored to treat EBV associated malignancies.

## 7. Conclusions and Outlook

Even though EBV specific T cell transfer, to treat virus associated malignancies, was one of the first cellular therapies (more than 25 years ago [91]), we still have very limited knowledge as to which T cell specificities and entities mediate this protective effect. While EBNA1, LMP1 and LMP2 have emerged as suitable targets and have even been clinically tested [54,55], the recent realization that early lytic EBV replication contributes to virus associated tumorigenesis provides additional antigens that could be explored [7]. More than one EBV antigen might also be necessary for immunotherapy, because any given T cell specificity seems to only target a subset of virus transformed cells at any given time. Redirecting T cells by transgenic TCR expression to recognize EBV antigens should be better tolerated during treatment of lymphomas and leukemias than eliminating the whole B cell compartment with CARs [81]. Therefore, similar as for adoptive T cell transfer therapy, EBV associated malignancies could also pave the way for TCR transgenic T cell therapy, by targeting several viral antigens.

## Figures and Tables

**Figure 1 cells-09-01400-f001:**
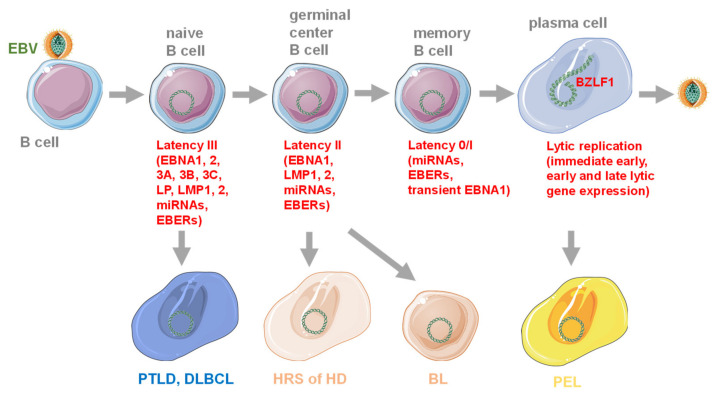
Epstein–Barr (EBV) associated B cell lymphomas emerge from different stages of EBV infection. Latency III with the indicated latent viral gene expression can be found in naïve B cells of healthy virus carriers, from which post-transplant lymphoproliferative disease (PTLD) and diffuse large B cell lymphoma (DLBCL) are thought to emerge. Reduced latency II viral gene expression is found in germinal center B cells, giving rise to Hodgkin-Reed-Sternberg (HRS) cells in Hodgkin’s disease (HD), as well as Burkitt’s lymphoma, with further down-regulation of LMP1 and 2. EBV persists in memory B cells without viral protein expression (latency 0) or transient EBNA1 expression (latency I), during homeostatic proliferation. Lytic EBV replication occurs after plasma cell differentiation from this persistence pool. The immediate early lytic transactivator BZLF1 kicks-off infectious virus particle production with immediate early, early and late lytic viral gene expression. Primary effusion lymphoma (PEL) is a plasmacytoma with elevated lytic EBV replication compared to other virus associated lymphomas. This figure was created in part with modified Servier Medical Art templates, which are licensed under a Creative Commons Attribution 3.0 unported license: https://smart.servier.com.

**Figure 2 cells-09-01400-f002:**
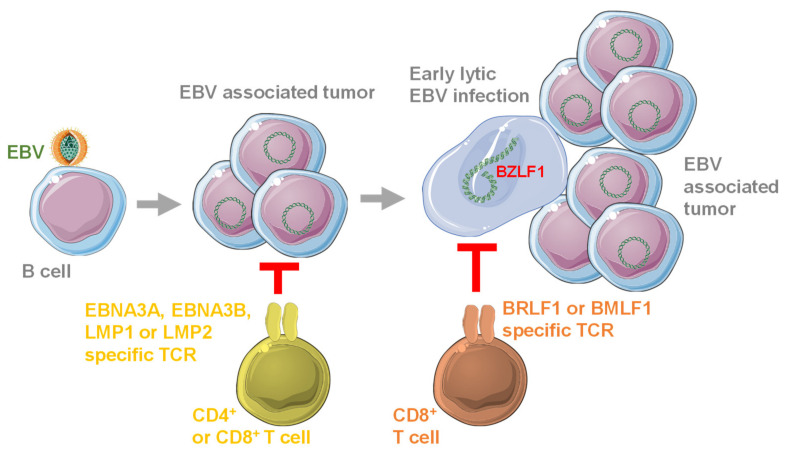
Available EBV specific T cell receptors (TCRs) target different stages of EBV infection and associated tumorigenesis. EBV transforms B cells after infection, by establishing latent viral gene expression from circularized viral genomes. This transforming latent EBV infection is targeted by TCR recognizing HLA presented EBNA3A, EBNA3B, LMP1 and LMP2. For further tumor growth and presumably tumor microenvironment conditioning, early lytic EBV infection in a subset of infected cells is required. This early lytic EBV gene expression is targeted by BRLF1 and BMLF1 specific TCRs. This figure was created in part with modified Servier Medical Art templates, which are licensed under a Creative Commons Attribution 3.0 unported license: https://smart.servier.com.

**Figure 3 cells-09-01400-f003:**
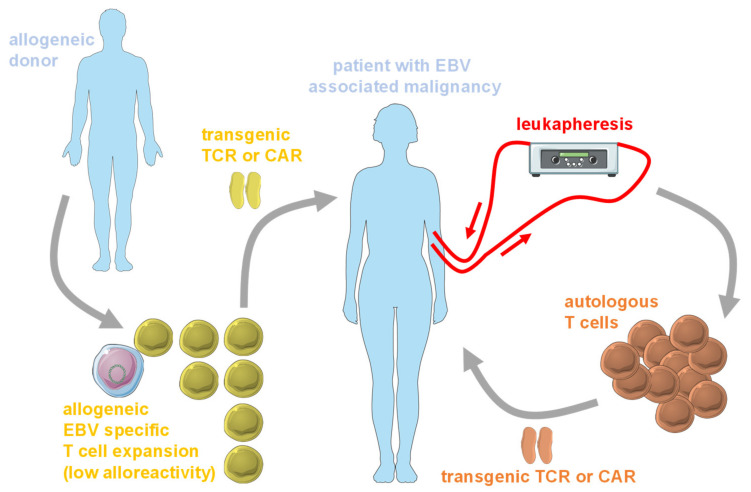
In addition to autologous T cells that can be obtained from leukapheresis of the peripheral blood of patients with EBV associated malignancies at sufficient numbers for TCR or chimeric antigen receptors (CAR) transgenic T cell therapies, allogeneic T cells can be used, but their alloreactivity against the recipient needs to be limited. EBV specific T cells that can be expanded from allogeneic T cells have been found to carry only low alloreactivity and are used for clinical trials with CAR transgenic T cells. This figure was created in part with modified Servier Medical Art templates, which are licensed under a Creative Commons Attribution 3.0 unported license: https://smart.servier.com.

**Table 1 cells-09-01400-t001:** Selection of clinical trials with transgenic T cell treatments targeting EBV associated malignancies from ClinicalTrials.gov (www.clinicaltrials.gov), that clearly name EBV positive tumors as one of their indications for enrollment.

Study Name	Conditions	Interventions	Trial Phase	Locations
**TCR transgenic T cells**				
EBV-TCR-T Cells for EB Virus Infection After HSCT (***NCT04156217***)	PTLD	HLA-A2, -A11 and -A24 restricted TCRs expressed in allogeneic donor T cells	Phase 1	Hebei (China)
EBV-TCR-T (YT-E001) for Patients With EBV-positive Recurrent or Metastatic NPC (***NCT03648697***)	NPC	HLA-A2, -A11 and -A24 restricted LMP1, LMP2 and EBNA1 specific TCRs expressed in autologous T cells	Phase 1/2	Fujian (China)
Phase I Trial of LMP2 Antigen-specific TCR T-cell Therapy for Recurrent and Metastatic NPC Patients (***NCT03925896***)	NPC	HLA-A2, -A11 and -A24 restricted LMP2 specific TCRs expressed in autologous T cells	Phase 1	Guangzhou (China)
**CAR transgenic T cells**				
In Vitro Expanded Allogeneic Epstein–Barr Virus Specific Cytotoxic T-Lymphocytes (EBV-CTLs) Genetically Targeted to the CD19 Antigen in B-cell Malignancies (***NCT01430390***)	ALL, Lymphoma	CD19 specific CAR expressed in allogeneic EBV specific T cells	Phase 1	New York (USA)
EBV CTLs Expressing CD30 Chimeric Receptors For CD30^+^ Lymphoma (CARCD30) (***NCT01192464***)	HD, NHL	CD30 specific CAR expressed in autologous EBV specific T cells	Phase 1	Houston (USA)
A New EBV Related Technologies of T Cells in Treating Malignant Tumors and Clinical Application (***NCT02980315***)	NPC	LMP1 specific CAR expressed in autologous T cells	Phase 1/2	Nanjing (China)
T-cells or EBV Specific CTLs, Advanced B-Cell NHL and CLL (ATECRAB) (***NCT00709033***)	CLL, NHL	CD19 specific CAR expressed in autologous EBV specific T cells	Phase 1	Houston (USA)
Allogeneic CD30.CAR-EBVSTs in Patients With Relapsed or Refractory CD30-Positive Lymphomas (***NCT04288726***)	ENKTL, HD. PTLD	CD30 specific CAR expressed in allogeneic EBV specific T cells	Phase 1	Houston (USA)
CD19-CAR Immunotherapy for Childhood Acute Lymphoblastic Leukemia (ALL) (CD19TPALL) (***NCT04288726***)	ALL	CD19 specific CAR expressed in EBV specific allogeneic donor T cells	Phase 1/2	Essen, Hannover, Frankfurt, Münster (Germany), Bristol, London (UK)
Phase I CD19/CD22 Chimeric Antigen Receptor (CAR) T Cells in Adults With Recurrent/Refractory B Cell Malignancies (***NCT03233854***)	DLBCL, ALL	CD19/CD22 specific CAR expressed in autologous T cells	Phase 1	Palo Alto (USA)
CD19 CAR and PD-1 Knockout Engineered T Cells for CD19 Positive Malignant B-cell Derived Leukemia and Lymphoma (***NCT03298828***)	ALL, BL	CD19 specific CAR expressed on autologous PD-1 knock-out T cells	Phase 1	Chongqing (China)
CARPALL: Immunotherapy With CD19 CAR T-cells for CD19+ Hematological Malignancies (***NCT02443831***)	ALL, BL	CD19 specific CAR expressed on autologous T cells	Phase 1	London, Manchester (UK)
Genetically Modified T-cell Infusion Following Peripheral Blood Stem Cell Transplant in Treating Patients With Recurrent or High-Risk Non-Hodgkin Lymphoma (***NCT01815749***)	NHL	CD19 specific CAR expressed in autologous T cells	Phase 1	Duarte (USA)
Cellular Immunotherapy Following Chemotherapy in Treating Patients With Recurrent Non-Hodgkin Lymphomas, Chronic Lymphocytic Leukemia or B-Cell Prolymphocytic Leukemia (***NCT02153580***)	NHL, CLL	CD19 specific CAR expressed in autologous T cells	Phase 1	Duarte (USA)

Abbreviations: PTLD, post-transplant lymphoproliferative disease; NPC, nasopharyngeal carcinoma; ALL, acute lymphoblastic leukemia; HD, Hodgkin’s disease; NHL, non-Hodgkin lymphoma; CLL, chronic lymphocytic leukemia; ENKTL, extranodal NK/T cell lymphoma; DLBLC, diffuse large B cell lymphoma; BL, Burkitt’s lymphoma.

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
