# Peer review of "Redirecting T Cells against Epstein–Barr Virus Infection and Associated Oncogenesis"

_cells, 2020, doi:10.3390/cells9061400_

Round 1

Reviewer 1 Report

In his review, Dr Munz addresses an important issue, i.e. T cell therapies for EBV-transformed malignancies. Important literature and relevant antigens are included.
However, I recommend a Major Revision as follows:
1) The author should make a distinction of primarily EBV-associated tumors vs. EBV-driven post-transplant lymphoproliferative disorder (PTLD). To this end, the manuscript will need Reorganization and also an additional paragraph.

2) The manuscript should also be revised to make a clear distinction between TCR and CAR T cell therapies.

3) In this context, the work by Theobald et al. in the group around R. Stripecke should be cited.

4) EBV-associated tumors proliferate within days and weeks, post-transplant lymphoproliferative disorder (PTLD) even within hours. A current vein-to-vein production of CAR T cells lasts four weeks, of TCR T cells even 6-8 weeks. What is the author’s strategy to overcome this therapeutical dilemma? This is a very hot topic which should be discussed in an additional paragraph which should be also accompanied by a graphical work.

5) Last but not least, the author should add a table listing closed and currently on-going trials with CAR and TCR T cell therapies targeting EBV. The size should not exceed one page, constitute a sound selection of a limited number of trials. Salient findings should be commented in brief in a right-handed column.

These 5 actions will majorly improve the quality and visibility / citation of this review.

Author Response

In his review, Dr Munz addresses an important issue, i.e. T cell therapies for EBV-transformed malignancies. Important literature and relevant antigens are included.

However, I recommend a Major Revision as follows:

1) The author should make a distinction of primarily EBV-associated tumors vs. EBV-driven post-transplant lymphoproliferative disorder (PTLD). To this end, the manuscript will need Reorganization and also an additional paragraph.

I have now added a new paragraph on pages 2 and 3 of the revised manuscript text that high-light the differences in PTLD and other EBV associated lymphomas, both with respect to time of occurrence as well as degree of immune suppression.

2) The manuscript should also be revised to make a clear distinction between TCR and CAR T cell therapies.

I had already discussed TCR and CAR T cell therapies in two distinct paragrahs and have now further high-lighted this distinction in a new clinical paragraph on pages 7 to 9 of the revised manuscript text.

3) In this context, the work by Theobald et al. in the group around R. Stripecke should be cited.

This work seems to discuss cytomegalovirus (HCMV) specific CAR T cell therapy that seems to be outside the scope of this review that focusses on EBV specific TCR and CAR transgenic T cell therapy. However, I have now cited on page 3 of the revised manuscript a study by the same group that discusses the phenotype of EBV specific T cells (Danisch et al., Am J Pathol 2019; new reference 43).

4) EBV-associated tumors proliferate within days and weeks, post-transplant lymphoproliferative disorder (PTLD) even within hours. A current vein-to-vein production of CAR T cells lasts four weeks, of TCR T cells even 6-8 weeks. What is the author’s strategy to overcome this therapeutical dilemma? This is a very hot topic which should be discussed in an additional paragraph which should be also accompanied by a graphical work.

I have now added a new clinical paragraph (section 6.) to the manuscript that discusses clinical trials registered in ClinicalTrials.gov that target EBV associated tumors. I have focused on two strategies: 1. Leukapheresis with direct transduction of autologous T cells and 2. Expansion of allogeneic EBV specific T cells as of the shelf product for transduction with reduced alloreactivity. These two approaches are further illustrated in the new figure 3. This paragraph could be expanded and form a review in its own right with biobanks of clonal T cell populations and NK cells as platforms for transgenic TCR and CAR expression being additional approaches. There are also more than 250 registered clinical trials with CD19 specific CARS. However, I felt that the scope of this review should be kept focused and therefore decided to not further expand this discussion.

5) Last but not least, the author should add a table listing closed and currently on-going trials with CAR and TCR T cell therapies targeting EBV. The size should not exceed one page, constitute a sound selection of a limited number of trials. Salient findings should be commented in brief in a right-handed column.

I have now added a new table 1 to list clinical trials with EBV specific TCRs and CARs that are registered in ClinicalTrials.gov. I had refrained from discussing these initially, because very little results are available yet. Accordingly, I have only discussed their design on page 9 of the revised manuscript text.

These 5 actions will majorly improve the quality and visibility / citation of this review.

Reviewer 2 Report

The author presents a review on adoptive cell therapy with T cells engineered to target Epstein Barr virus infected cells. In this context, the benefit of strategies utilizing a TCR specific for EBV early lytic antigens or transforming latent antigens in contrast to a CAR targeting B cell antigens is highlighted.

This is a comprehensive and well-written review. I have some suggestions as outlined below.

1. The latency stages, expression of EB viral antigens, stages of maturation of infected B cells and the corresponding malignant cells should be visualized in a graphical abstract to help the non-specialist reader to recapitulate the potential targets for redirected therapy. Such a figure may potentially be integrated into or associated to Fig.1

2. lines 71/72...” and molecules that are identified by these primary immunodeficiencies during EBV specific immune control” should read ...” identified by these primary immune cells during ...”, please check or re-phrase. Lines 114/115 should read “However, CD4+ and CD8+ T cells in even higher frequencies ... recognize ...” Rephrase line 179 “...is maybe only slightly lower...”, also lines 221/222. Please carefully check phrasing throughout the manuscript.

3. Given a strong rational for EBV specific TCR T cell therapy, the reason(s) for failure so far and the strategies for the future should be more precisely discussed: how to increase efficacy in vivo, how to improve killing, which early lytic antigens are favorite targets and which should be targeted in combination, how is tumor heterogeneity addressed etc

Author Response

The author presents a review on adoptive cell therapy with T cells engineered to target Epstein Barr virus infected cells. In this context, the benefit of strategies utilizing a TCR specific for EBV early lytic antigens or transforming latent antigens in contrast to a CAR targeting B cell antigens is highlighted.

This is a comprehensive and well-written review. I have some suggestions as outlined below.

  1. The latency stages, expression of EB viral antigens, stages of maturation of infected B cells and the corresponding malignant cells should be visualized in a graphical abstract to help the non-specialist reader to recapitulate the potential targets for redirected therapy. Such a figure may potentially be integrated into or associated to Fig.1

I have now added a new figure 1 to high-light the different EBV gene expression patterns in B cell differentiation stages and the EBV associated lymphomas that are thought to emerge from these.

  1. lines 71/72...” and molecules that are identified by these primary immunodeficiencies during EBV specific immune control” should read ...” identified by these primary immune cells during ...”, please check or re-phrase. Lines 114/115 should read “However, CD4+ and CD8+ T cells in even higher frequencies ... recognize ...” Rephrase line 179 “...is maybe only slightly lower...”, also lines 221/222. Please carefully check phrasing throughout the manuscript.

I have now rephrased all of these sentences.

  1. Given a strong rational for EBV specific TCR T cell therapy, the reason(s) for failure so far and the strategies for the future should be more precisely discussed: how to increase efficacy in vivo, how to improve killing, which early lytic antigens are favorite targets and which should be targeted in combination, how is tumor heterogeneity addressed etc

I have now added a new clinical paragraph (section 6.) to discuss the registered clinical trials to target EBV associated malignancies. A selection of these is also summarized in the new table 1. These suggest that it is just early days and in light of the success of adoptive EBV specific T cell transfer there is in my opinion no reason to think that TCR transgenic T cells will not be similarly efficient as enriched naturally occurring EBV specific T cells in the treatment of EBV associated malignancies.

Reviewer 3 Report

This review provides an accurate and concise summary of current knowledge on T cell control of EBV infection and how this is being applied to the development of T cell based therapies for EBV-associated cancers. I have no major concerns with the review. A few minor points are listed below:

  1. Line 10 : To avoid any possible confusion, when referring to “EBV specific autologous T cell”, it may be helpful to refer to them as naturally occurring to make a clear distinction between these and T cells that have been engineered by TCR gene transfer (which are also EBV-specific and autologous).
  2. Line 42 : it is stated that in Burkitt’s lymphoma only EBNA1 is expressed as the sole viral protein. This is generally correct, but 10–15% of endemic Burkitt's lymphoma cases express EBNA1, EBNA3A, EBNA3B, EBNA3C, EBNA-LP and BHRF1 (Kelly GL, et al. 2009. PLOS Pathog. 5: e1000341). It would be more accurate to say that the majority of Burkitt's lymphoma cases only express EBNA1
  3. Line 52 and 53 : state: “For example, nasopharyngeal carcinoma (NPC) expresses often latency II and EBV associated gastric carcinoma sometimes even latency I [19]”. However reference 19 does not mention the latency state of EBV in gastric cancer
  4. Some minor typographical errors need correcting (eg lines 17, 21, 118)

Author Response

This review provides an accurate and concise summary of current knowledge on T cell control of EBV infection and how this is being applied to the development of T cell based therapies for EBV-associated cancers. I have no major concerns with the review. A few minor points are listed below:

Line 10 : To avoid any possible confusion, when referring to “EBV specific autologous T cell”, it may be helpful to refer to them as naturally occurring to make a clear distinction between these and T cells that have been engineered by TCR gene transfer (which are also EBV-specific and autologous).

I now revised the abstract to include naturally occurring for in vivo primed EBV specific autologous T cells.

Line 42 : it is stated that in Burkitt’s lymphoma only EBNA1 is expressed as the sole viral protein. This is generally correct, but 10–15% of endemic Burkitt's lymphoma cases express EBNA1, EBNA3A, EBNA3B, EBNA3C, EBNA-LP and BHRF1 (Kelly GL, et al. 2009. PLOS Pathog. 5: e1000341). It would be more accurate to say that the majority of Burkitt's lymphoma cases only express EBNA1

I have now indicated on page 1 of the revised manuscript text that most Burkitt’s lymphomas are latency I.

Line 52 and 53 : state: “For example, nasopharyngeal carcinoma (NPC) expresses often latency II and EBV associated gastric carcinoma sometimes even latency I [19]”. However, reference 19 does not mention the latency state of EBV in gastric cancer

I have now added on page 2 of the revised manuscript version the new reference 20 (Shannon-Lowe and Rickinson, Front Oncol 2019) that includes the latency state of EBV in gastric carcinoma.

Some minor typographical errors need correcting (eg lines 17, 21, 118)

These have been corrected.

Round 2

Reviewer 1 Report

The author followed all recommendations of the Reviewer. Therefore I recommend the work for publication now.